# HCAP: Hybrid cyber attack prediction model for securing healthcare applications

**Mohanad Faeq Ali[1], Mohammed Shakir Mohmood[2], Ban Salman Shukur[3], Rex Bacarra[4], Jamil Abedalrahim Jamil Alsayaydeh**[ID][5]*, **Masrullizam Mat Ibrahim[5], Safarudin Gazali Herawan**[ID][6]

**1** Fakulti Teknologi Maklumat dan Komunikasi, Universiti Teknikal Malaysia Melaka (UTeM), Melaka, Malaysia, **2** Scholarship and Cultural Relations Directorate,Ministry of Higher Education and Scientific Research, Baghdad, Iraq, **3** Computer Engineering Techniques Department, Technical Engineering College, Al-Bayan University, Baghdad, Iraq, **4** Department of General Education and Foundation, Rabdan Academy, Abu Dhabi, United Arab Emirates, **5** Department of Engineering Technology, Fakulti Teknologi Dan Kejuruteraan Elektronik Dan Komputer (FTKEK), Universiti Teknikal Malaysia Melaka (UTeM), Melaka, Malaysia, **6** Industrial Engineering Department, Faculty of Engineering, Bina Nusantara University, Jakarta, Indonesia

* jamil@utem.edu.my

**Editor:** mamoona humayun, University of Roehampton, UNITED KINGDOM OF GREAT BRITAIN AND NORTHERN IRELAND

## Abstract

The rapid development and integration of interconnected healthcare devices and communication networks within the Internet of Medical Things (IoMT) have significantly enhanced healthcare services. However, this growth has also introduced new vulnerabilities, increasing the risk of cybersecurity attacks. These attacks threaten the confidentiality, integrity, and availability of sensitive healthcare data, raising concerns about the reliability of IoMT infrastructure. Addressing these challenges requires advanced cybersecurity measures to protect the dynamic IoMT ecosystem from evolving threats. This research focuses on enhancing cyberattack prediction and prevention in IoMT environments through innovative Machine-learning techniques to improve healthcare data security and resilience. However, the existing model's efficiency depends on the diversity of data, which leads to computational complexity issues. In addition, the conventional model faces overfitting issues in training data, which causes prediction inaccuracies. Thus, the research introduces the hybridized cyber attack prediction model (HCAP) and analyzes various IoMT data source information to address the limitations of dataset availability issues. The gathered information is processed with the help of Principal Component-Recursive Feature Elimination (PC-RFE), which eliminates the irrelevant features. The extracted features are fed into the lion-optimization technique to fine-tune the hyperparameters of the recurrent neural networks, enhancing the model's ability to efficiently predict cybersecurity threats with a maximum recognition rate in IoMT environments. The recurrent networks, specifically Long Short Term Memory (LSTM), process data from healthcare devices, identifying abnormal patterns that indicate potential cyberattacks over time. The created system was implemented using Python, and various metrics, including

**Data availability statement:** All the datasets used in this study are available from the Zenodo database (https://zenodo.org/records/14059543).

**Funding:** The author(s) received no specific funding for this work.

**Competing interests:** NO authors have competing interests

false positive and false negative rates, accuracy, precision, recall, and computational efficiency, were used for evaluation. The results demonstrated that the proposed HCAP model achieved 98% accuracy in detecting cyberattacks and outperformed existing models, reducing the false positive rate by 25%. The false negative rate by 20% and a 30% improvement in computational efficiency enhances the reliability of IoMT threat detection in healthcare applications.

## Introduction

The IoMT has transformed healthcare through the integration of medical devices and networks of communication with necessary protocols, delivering many benefits while raising cybersecurity risks [1]. As IoMT devices grow more important in healthcare, protecting sensitive health data integrity is imperative [2]. Cyberattacks on IoMT devices can compromise patient data, disrupt medical services, and manipulate device operations, threatening healthcare data [3]. Concerns over the quality of security and privacy in healthcare data have precluded widespread usage of IoMT datasets, making security modelling challenging and computationally expensive [4]. Due to IoMT integration, cyberattacks on healthcare systems have proliferated, offering serious threats to patients and providers, like data breaches and medical device manipulation [5].

Conventional security measures often fall behind these threats where IoMT devices in the healthcare service raise security and privacy concerns for critical health data. Therefore, IoMT setups need innovative prediction and detection techniques to recognize and prevent cyber attacks [6] proactively. Cyberattacks in the healthcare sector affect the privacy and security of patient data due to the interrelated IoMT devices [7]. In IoMT applications, robust authentication techniques are essential for maintaining secure communication. A service-reliant authentication approach has proven effective in enhancing the resilience of IoT-powered systems, particularly in smart cities. This approach can be adapted to bolster security in healthcare IoT environments, ensuring robust access control and data protection [8]. Existing research highlights that traditional network attack prediction techniques are inadequate for securing IoMT, necessitating more sophisticated ML-based approaches [9]. Artificial Intelligence (AI) technologies called ML are employed to identify cyber-attacks and ensure the security of IoMT devices in the healthcare system [10] to address security concerns. In healthcare, signature-based and anomaly-based detection systems are commonly used to identify cyber threats and often suffer from high FPR. Applying them to varied and massive IoMT datasets reduces their efficiency [11]. While ML offers a range of methods and solutions to enhance cyberattack recognition, mitigation, and prevention, the most recent contributions to state-of-the-art benchmark datasets fail to address significant features [12].

ML techniques have emerged as a powerful tool for detecting and predicting cyber threats in healthcare applications to address these security concerns. However, traditional ML models face computational complexity challenges [13]. Thus,

the research discussed the possible threats and recent security issues in healthcare. It provides a novel cyber attack classification and its impact on the functioning of integrated IoMT devices in the healthcare system [14]. The growing use of these devices makes the system open to cyber attacks, which include Denial of Service (DoS) and data theft via eavesdropping and other phishing attacks [15]. Due to the dynamic nature of IoMT and its unnecessary greater frequency of retraining the features leads to prediction inaccuracy [16]. The communication frequency from the IoMT device with varying time intervals monitors the corresponding affected devices and optimizes the transmission of fast data rate [17]. A nature-inspired optimization technique is performed to fine-tune the recurrent neural networks [18] to predict cyber-attacks in healthcare data accurately. To predict the types of cyber-attack events in healthcare data, use Long Short-Term Memory (LSTM) recurrent networks [19]. Increased cybersecurity attacks from rapid IoMT device integration create new vulnerabilities and data complexities, motivating this research's proposed use of the HCAP model. Since conventional security solutions struggle with real-time detection and IoMT data patterns like communication frequencies, protocols, and device conditions, this diversity of networked devices requires efficient security measures. Predicting cyber-attacks in healthcare systems is essential for mitigating risks. In a similar approach, Kovtun et al. [20] developed an entropy-extreme model for early-stage cyber epidemic prediction, which informs our focus on temporal analysis in IoMT systems. HCAP uses advanced ML approaches, such as PC-RFE, to extract features that improve threat identification, and for better prediction, lion-optimized LSTM networks are applied to achieve better computational processing efficiency.

Research Gap: Traditional cyber attack detection models have several flaws well-documented in the literature. These include inefficient processing of high-dimensional security data, use of suboptimal feature selection methods, lack of adaptability to new attack vectors, and high rates of false positives and false negatives. By using Principal Component-Recursive Feature deletion (PC-RFE) for dimensionality reduction and optimum feature selection, we guarantee the deletion of redundant or non-informative features in the Hybridized Cyber Attack Prediction Model (HCAP), which we propose to solve these shortcomings. Cyber assault patterns may be robustly sequentially modelled using Long Short-Term Memory (LSTM) networks, which successfully capture temporal relationships in data from network traffic.

Novelty: To optimize feature selection and sequential threat prediction in IoMT environments, this study developed the Hybridized Cyber Attack Prediction Model (HCAP). PC-RFE stands for Principal Component-Recursive Feature Elimination, and LSTM stands for Long Short-Term Memory. By utilizing Principal Component Analysis (PCA) to decrease dimensionality and recursively eliminate redundant or non-informative features, PC-RFE improves computational efficiency compared to conventional approaches. This guarantees the selection of highly discriminative attributes for cyber attack detection. By including LSTM, the model can better identify anomalies in network traffic and security logs by capturing long-range dependencies, a major improvement over typical machine learning (ML) classifiers, which have difficulty handling sequential relationships. Furthermore, the hybrid method improves detection accuracy and reliability by decreasing the False Positive Rate (FPR) and the False Negative Rate (FNR). With limited resources, HCAP guarantees strong access control, accurate threat prediction in real-time, and extensible cybersecurity solutions. It is a flexible solution for cyber assault mitigation in high-risk, ever-changing digital ecosystems, and its applicability goes beyond smart cities and industrial IoT.

The prime contributions of this research include the following aspects:

1) To develop a Hybridized Cyber Attack Prediction (HCAP) model that uses lion-optimized LSTM recurrent neural networks to detect complex IoMT attack patterns across multiple communication protocols.

2) To select important features, reduce dimensionality, improve accuracy, and remove irrelevant features combined, the PC-RFE technique is applied.

3) Enhance attack prediction and computational efficiency by providing selected features into LSTM networks, analyzing the temporal dependencies, and fine-tuning hyperparameters using the lion optimization algorithm.

4) To accurately predict regular events and cyberattacks in real-time, with minimal false positives and relatives, thus improving healthcare data security.

The research article is organized as follows: Section 2 contains a brief literature review of cyberattacks performed in healthcare applications based on IoMT. Section 3 discusses the development of the HCAP model for analyzing cyberattacks in healthcare scenarios using the lion-optimized recurrent learning model. Section 4 performs an experimental analysis of the CICIoMT dataset with its associated attack types, and Section 5 concludes the paper, providing future research scope and implications.

## Literature review

Due to privacy concerns and the lack of diverse datasets for detecting cyberattacks, Ahmed et al. [21] applied nearest-neighbor-based algorithms to analyze attack behaviour and provide corresponding countermeasures. The study examines various sensors and medical devices that can exploit IoT devices in a healthcare application. This research identified anomalies and focused on scaling DoS and injection attacks.

Applying AdaBoost for classification and Particle Swarm Optimization (PSO) for feature selection, Sun et al. [22] presented a hybrid PSO-AdaBoost IDS for intelligent healthcare devices, which was tested on the NSL-KDD dataset. It detects many kinds of attacks, such as DoS, U2R, R2L, and Probe, with improved accuracy and recalls; AdaBoost demonstrates the highest recall at 0.96. Further validation in real-world healthcare environments was necessary to ensure the system's scalability and reliability; however, it does improve IoMT security and patient data protection.

Zubair et al. [23] addressed the issue of Bluetooth-based cyber-attacks by proposing a decentralized intrusion detection solution for competent healthcare IoMT networks based on deep learning. The multilayer detection models achieved F1 scores of 97–99.5% using the BlueTack dataset, which encompasses Bluetooth Classic and BLE attack data. While the system's effectiveness in real-time and different IoMT scenarios needs additional confirmation, key features include excellent accuracy in detecting and decentralized edge network deployment.

For efficient cyber-attack and anomaly detection in the IoMT in healthcare applications, they were suggested by Kilincer et al. [24] to merge-recursive feature elimination (RFE) with a multilayer perceptron (MLP). Several datasets, such as ECU-IoHT, ICU, and Telemetry Operating and Network (TON) based IoT, have attained remarkable accuracy rates of up to 99.99%. The model shows strong performance and resilience against cyber attacks. Still, it may not be able to be applied to other IoMT contexts due to its dependence on hyperparameter adjustment and the risk of overfitting for specific datasets.

A privacy-enhanced IoMT system that was well-suited for virtual patient diagnosis and treatment was presented by Subramaniam et al. [25]. Clustering based on low energy consumption, environmental monitoring using circular assisted Hidden Markov Model (HMM), and Twine-LiteNet encryption were all-important methods. By analyzing the route selection with Search and Rescue Optimization (SRO), the technique boosts network performance, lifetime, energy efficiency, and latency/packet drop rate reduction. The result of network simulations demonstrated a 35% improvement in lifetime.

An ML-based solution to improve the safety of IoMT devices was suggested by Rani et al. [26] called SmartHealth Framework (SHF), which uses an Artificial Neural Network (ANN) which monitors vital signs and identifies inappropriate behaviour. It obtains a 92% detection accuracy with an F1-score of 90%, showcasing its exceptional performance in device tampering, DoS attacks, and three concurrent fake data injection attack scenarios. The result can distinguish between benign and severe attack actions by tracking the user's bodily processes in real time.

Kulshrestha, P., & Vijay Kumar [27] aim to improve the security of healthcare records by developing an Intrusion Detection System (IDS) that uses ML to detect cyberattacks on IoMT-based systems. After evaluating several ML classification algorithms, the most effective model was Adaptive Boosting (AB) according to metrics like accuracy, F1-score, and False Positive Rate (FPR). Balhareth, G., & Ilyas [28] suggested an IDS for IoMT networks that enhances detection efficiency

and accuracy using filter-based feature selection approaches, including XGBoost and Mutual Information (MI), in addition with tree-based ML classifiers. The model attained an accuracy of 98.79% and a False Alarm Rate (FAR) of 0.007 when tested on the CIC-IDS dataset. Although it works well for binary classification, it was tested in real-world IoMT scenarios and focuses on multi-class detection.

An intrusion detection system for IoMT and IoT networks was suggested by Mahato et al. [29] employing K-Nearest Neighbor (KNN) for classification and Chaotic Grey Wolf Optimizer (CGWO) for feature selection, with the addition of eleven chaotic maps. The generalizability of the results in network traffic-based attack detection provides relevant features and improves detection accuracy, depending on additional validation related to cyber threats in healthcare settings.

LAILA M. HALMAN et al. [30] suggested the Machine Learning Based Cyberattacks Detector (MCAD) in Software-Defined Networking (SDN) for Healthcare Systems. By modifying an L3 learning switch application to gather typical and out-of-the-ordinary traffic and deploying MCAD on the Ryu controller, our primary objective is to suggest a machine learning-based cyberattack detector (MCAD) for healthcare systems. The results may help reduce the severity of assaults on healthcare application security. With an F1-score of 0.9998 for the normal class and 0.9882 for the attack class, the MCAD demonstrates remarkable performance and high dependability. Throughput-wise, MCAD reached 5,709,692 samples/s, indicative of a complicated, high-performance real-time system. Furthermore, it improved the network's key performance indicators (KPIs), with throughput improving by 60.9% and latency and jitter reducing by 77% and 23%, respectively, compared to the attack outcomes.

Atheer Alaa Hammad [31] proposed the Random Forest and LSTM Hybrid Model for Detecting DDoS Attacks in Healthcare IoT Networks. The suggested hybrid RF-LSTM variant was tested on actual healthcare IoT datasets worldwide. RF achieved a precision of 93% and an accuracy of 94% when identifying static abnormalities. With a 91% F1 score and low false negatives, LSTM demonstrated exceptional performance while dealing with temporal dependencies. Using both algorithms together improved the device's detection accuracy to 97% in real-time circumstances, which is impressive given the wide range of assaults it can identify. Maintaining patient security and information integrity, this study emphasizes the capacity of hybrid designs to maintain the safety of the Internet of Things (IoT) and minimize cyber risks in healthcare contexts.

The research on IoMT cyberattack detection highlighted numerous significant obstacles. A lack of different datasets makes model training and validation difficult, potentially causing overfitting and lower efficacy when applied to fresh data. Many research investigations focus on certain attack types or healthcare scenarios, such as DoS or Bluetooth-based attacks, restricting their applicability to other healthcare contexts. Conventional ML methods complicate feature selection and optimize the diverse data source with varying model parameters, impacting model performance. Privacy-enhanced technologies designed to track patient monitoring health device data and ensure its security must be focused in healthcare. Thus, the necessary security model has to operate in real-time and varied IoMT environments for cyberattack detection and evaluation.

The current cybersecurity solutions discussed for IoMT devices in healthcare settings pose significant shortcomings. For example, PSO-AB-IDS achieved 96.6% accuracy and often failed to detect sophisticated attacks like zero-day vulnerabilities and risks exposing patient-related sensitive data. Likewise, models like HMM-SRO struggle harder to scale in more extensive healthcare settings, leaving the attacks unaddressed. SHF-ANN has a high false positive rate of around 15%, which can lead to fatigue and missing real attacks. Then, XGBoost were too complex when applied in real-time processing, which is more critical for emergency scenarios. Some other models also tend to face overfitting problems during the training phase that impact their model performance and lack of analyzing temporal data. KNN-CGWO struggles with large datasets due to heavy traffic in IoMT devices. This is essential for identifying abnormal communication patterns over time, and thus, it fails to detect cyberattacks effectively in healthcare settings. As mentioned earlier, these shortcomings highlight the need for better accuracy, computational efficiency, and robustness in cybersecurity solutions for healthcare environments.

## Hybridized cyber attack prediction model on healthcare application

The research aims to design a robust HCAP model to predict cyberattacks and improve healthcare data security within IoMT infrastructure systems. The model is built to manage the complex and diverse data produced by a wide range of IoMT devices that communicate over different protocols like Bluetooth, MQTT, and WiFi. Processing this varied data accurately while reducing computational complexity is a primary objective. Furthermore, the research intends to solve the prevalent problem of overfitting by applying optimization and feature selection methods, guaranteeing that the model may successfully generalize to previously unreported health data. The system's ultimate goal is to improve the security and reliability of IoMT devices in healthcare settings by achieving high attack identification rates and precisely recognizing cyberattacks and abnormal occurrences in real-time, with few false positives and negatives.

### Data collection from ciciomt2024 dataset

The data collection process involves a multi-protocol IoMT dataset [32] designed for benchmarking security solutions that include multiple cyberattack scenarios across different IoMT devices. The dataset contains information from 25 accurate and 15 simulated IoMT devices, comprises 40 devices in total, and covers a variety of communication protocols, like WiFi, MQTT, and Bluetooth, that are widely used in healthcare applications, as shown in Fig 1.

The data are taken from Major US health data breaches [33]. At least 500 people in the United States were impacted by health data breaches described in this dataset. Any breach involving the unlawful use or disclosure of protected health information must be reported by organizations and entities covered by the legislation, as mandated by the Health Insurance Portability and Accountability Act (HIPAA) in 1996. Name of Covered Entity, State, Type of Covered Entity, Date of Breach Submission, Type of Breach, Location of Breached Information, Present Business Associate, and Web Description are all mandatory variables in this dataset. From 2009 to December 2019, this dataset has over 860 items, offering a comprehensive look at the effects and consequences of these breaches on healthcare institutions and their patients. Everyone interested in healthcare security best practices and preventing future data-related mishaps may benefit from this data, not only healthcare professionals.

Cybersecurity issues stemming from managing different devices and protocols can be better understood with the help of visual representations that highlight the variety and complexity of IoMT setups in this healthcare data. The components listed in Fig 1 show how important it is to have a cybersecurity solution that is both strong and adaptable so that it can deal with the specific risks that come with various data sources and protocols.

Table 1 details the CICIoMT2024 dataset's information regarding IoMT devices, cyberattacks, and communication protocols in healthcare applications. Traffic from versatile pagers, baby monitors, emergency SOS controls, and security cameras is included in the dataset. It detects 18 attack categories, such as DDoS, DoS, reconnaissance, MQTT-based cyber attacks, and Spoofing, which shows IoMT settings' diverse risks. Data is gathered in.PCAP and CSV formats are used for packet evaluation and machine learning (ML) development of models. The integration of systems for security, as demonstrated in previous research [34], can also inform IoMT-based security measures for healthcare applications where

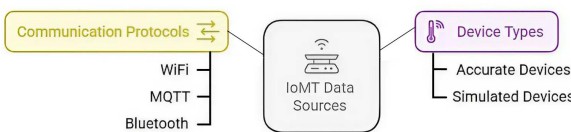

**Fig 1. Analysis of IoMT data sources.**

early prevention is crucial. This dataset is essential for testing the HCAP model's ability to identify and mitigate IoMT cyber threats.

Fig 2 shows the HCAP model's systematic flow from data gathering to attack prediction, providing a thorough overview. The preprocessing of data, extraction of features, and use of machine learning techniques are all visually represented. The graphical summary offers a concise overview of the model's methodology, highlighting its ability to handle various forms of IoMT data and produce reliable predictions about possible cyber assaults. The significance of the HCAP model

**Table 1. IoMT devices and attack overview in the CICIoMT2024 dataset.**

| Attribute | Details |
|---|---|
| Devices Included | Multifunctional Pager, Sense U Baby Monitor, SINGCALL SOS Button, Ecobee Camera, Blink Mini, Raspberry Pi 4, 34 additional IoMT devices |
| Types of Attacks | DDoS, DoS, Recon, MQTT, Spoofing, 13 other attack types |
| Protocols Used | WiFi, MQTT, Bluetooth |
| Profiling | Power, idle, active, interaction |
| Data Format | .PCAP, CSV |

## HCAP: Hybrid Cyber Attack Prediction Model

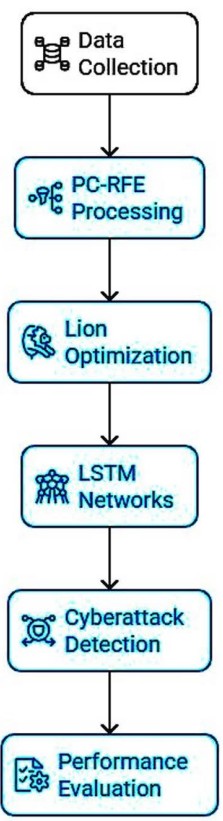

**Fig 2. Graphical abstract of HCAP.**

in improving cybersecurity in IoMT contexts is highlighted in this research, which also highlights its integrated nature of lion-optimized recurrent neural networks.

## Data preprocessing

In this phase, the data is categorized based on the communication protocols used as Bluetooth data is organized into benign and attack traffic, with the profiling data obtained from power experiments. Wireless Fidelity (WiFi) and Message Queuing Telemetry Transport (MQTT) data included in.pcap files and.csv files comprises the features with different attack categories of IoMT infrastructure with device states like power, idle, active, and interaction states. The communication patterns with a time series of these sequences and a burst of messages from a monitoring device can significantly affect a potential spoofing attack.

Efficient communication is essential in healthcare IoMT environments, just as in VANET systems for safety communications [35]. Such techniques can also be adapted to control traffic congestion in healthcare communication systems.

**Normalization.** Z-score normalization measures the standard deviations observed in Eq. (1).

$$Z = \frac{X - \mu}{\sigma}$$

(1)

Where $Z$ is the anomaly score identified in the cyberattack of IoMT devices, $X$ is the network traffic with packet rate feature and latency of packets with different lengths during the active or interaction experiments. The variable $\mu$ indicates the mean representation from power or idle states, and $\sigma$ represents the feature's standard deviation. This process allows the HCAP model to detect normal IoMT behaviour, aiding in attack detection from deviations.

## Feature extraction

Attack data in raw. pcap files need feature extraction. This technique extracts the length of the header data, protocol types, time frame, source/destination IP address, health care data with appropriate packet size, and Time To Live (TTL) from network traffic data. These feature records are crucial to identifying malicious network behaviour. Principal Component-based Recursive Feature Elimination (PC-RFE) decreases dimensionality by identifying the most relevant features while conserving data variation. PCA minimizes noise and improves data clarity for attack detection by filtering out less important components. When PCA is performed, the covariance matrix of the data is computed, and its eigenvalues are calculated. Each eigenvalue corresponds to a principal component and represents the variance captured by that component. The explained variance for each principal component is given by its eigenvalue, and the total variance of the data source is the sum of all eigenvalues. The proportion of variance explained by each component helps to guide the feature selection process.

$$Z = XW$$

(2)

Where $X \in \mathbb{R}^{n \times p}$ Eq. (2) is the input data matrix, where $n$ is the no.of observations and $p$ is the no.of features. The parameter $W \in \mathbb{R}^{p \times k}$ is the matrix of weights corresponding to the $k$ principal components, and $Z \in \mathbb{R}^{n \times k}$ is the transformed data in the new feature space called principal components. The feature extraction process from PCAP files is as follows: CSV files simplify the attack detection. The extracted vital features include 44 features which are the length of the header packet, lifetime of the packet during transmission, speed of packet transmission within a flow, IP addresses of the source and destination, length of the packet as minimum and maximum, and the TTL, protocol types like Transmission Control Protocol (TCP), User Datagram Protocol (UDP), Internet Control Message Protocol (ICMP) with the associated TCP flags like SYN, ACK, FIN, RST indicating the state of the connection. Recursive Feature Elimination (RFE) eliminates the less important features to improve the model's generalization capabilities. Effective network resource control is vital for

managing IoMT devices across various protocols. Managing 5G clusters in critical infrastructure is a strong example of how optimizing network resources is essential [36], aligning with the specific needs of IoMT systems in healthcare environments. By prioritizing features according to their value, RFE improves the HCAP model's accuracy by retaining the most predictive features.

$$Rank\,(f_i) = \frac{\partial L(y, f_i(X))}{\partial f_i} \qquad (3)$$

Wherein Eq. (3), $L$ indicates the loss function, $y$ represents the target variable, and $f_i(X)$ represents the prediction function for feature $i$. An iterative process is used to delete features until a certain number of features are left, with features prioritized according to their importance scores. Features are ordered for the ranking process, where the lowest-ranked ones are eliminated iteratively, which helps to fix overfitting problems and eliminates redundant features.

### Feature set analysis and reduction using lion-optimized recurrent networks

Model training uses the most relevant and informative features after feature extraction. Inspired by lion hunting, Lion Optimization chooses features to improve model accuracy in predicting cyber attacks. Optimization automatically selects the most important data items to improve the prediction model's efficiency and efficacy. This process reduces noise and irrelevant data by retaining only features that help to predict threats. The LSTM-based lion-optimized recurrent network proceeds to the HCAP model, where integrating a lion-optimization algorithm into this recurrent neural network variation improves cyber threat prediction. A fully connected dense layer is used for classification after three LSTM layers with different numbers of neurons (128, 64, and 32, respectively) in the model. Using the Adam optimizer, this study tuned the learning rate to 0.001 to achieve a compromise between stability and convergence speed. Set the dropout rate to 0.3 to avoid overfitting and keep the model generalizable. To provide reliable training updates without significant memory overhead, the batch size is defined as 64. The hidden layers are treated using ReLU (Rectified Linear Unit) to address non-linearity. The output layer employs Sigmoid for binary classification, and for multi-class attack detection, it employs Softmax.

**Optimizing weights and hyperparameters.** Nature inspires the lion optimization algorithm, which inspires social behaviour and territorial movements of lions, and it effectively balances the exploration that can find new solutions and exploitation that refines optimal solutions for predicting cyberattacks identified in IoMT devices integrated into healthcare applications.

Initialize Lions (Solutions) Population: Each lion is initialized with a set of populations that represents a set of RNN hyperparameters, that is, the learning rate $lr_i$ ranges from 0.001 to 0.1, the number of hidden units as $hu_i$ ranges between 50–200, and the dropout rate is termed as $dr_i$ with the range from 0.2 to 0.5. Any parameter that exceeds its boundary is reset to the nearest boundary value.

$$X_i = [lr_i, hu_i, dr_i] \qquad (4)$$

In Eq. (4), the parameter $X_i$ represents the solution of lion $i$ with the set of RNN hyperparameters, and each hyperparameter is initialized within the bounds $rand(a, b)$ provided.

Assign Lions as Pride and Nomad Lions: The population is divided into pride lions who exploit known attack areas and nomad lions called to explore new unknown attack areas. The pride lions aim to fine-tune hyperparameters that improve the RNN's performance on identified attack types in the CICIoMT dataset, such as DoS, DDoS, and spoofing attacks. These data sources emphasize refining solutions through minor and local adjustments. The pride lions focus on local improvements, while the nomad lions search for global solutions. They focus on global search, exploring radical changes in hyperparameters, aiming to improve the RNN's ability to detect previously unknown attack patterns or abnormal behaviour that may arise in healthcare IoMT devices.

Lion Mating: Lions combine solutions through crossover-like mechanisms, which allows the algorithm to explore diverse regions of the parameter space for pride lions. These pride lions occasionally mate, combining their hyperparameters with other lions within their pride to create new solutions. The process mimics crossover in genetic algorithms and promotes diversity within the pride. Mating introduces diversity into the pool of RNN hyperparameters, allowing the detection of the model to consider a broader set of configurations for improved accuracy in detecting various attack types. For two lions, $X_1 = [lr_1, hu_1, dr_1]$ and $X_2 = [lr_2, hu_2, dr_2]$ the crossover Eq. (5) is given as:

$$X_{offspring} = \alpha X_1 + (1 - \alpha)X_2$$

(5)

Where $X_{offspring}$ indicates a new lion solution, $\alpha$ for random numbers considered as 0 and 1. The term $X_1$ and $X_2$ indicates the parent's solutions and generates a new lion by combining the hyperparameters of these $X_1$ and $X_2$. This crossover mechanism ensures that different combinations of learning rates, hidden units, and dropout rates are explored, further refining the ability of the RNN to detect complex cyberattacks. Exploration and exploitation are both balanced in the lion optimization method. Exploration involves searching new parts of the parameter space to identify cyber-attacks, while exploitation involves refining known suitable regions.

Fig 3 shows the HCAP model optimizes RNN hyperparameters for IoMT cyber attack detection using the lion optimization technique. Initializing a lion population with unique hyperparameters starts it. Then, pride lions refine local solutions, and nomad lions explore the more expansive parameter space. The method determines each lion's fitness by detecting attacks with the RNN, then mating and territorial defence to combine and improve hyperparameters. Update the best-known solution until a termination condition is reached by evaluating new solutions. A method that balances exploration and exploitation allows the HCAP model to adapt to new cyber threats while preserving computational efficiency, boosting its generalization across attack types and IoMT devices. It is crucial to have a convergence rate that allows the algorithm to explore effectively while swiftly settling on ideal parameters to adapt to the ever-changing nature of IoMT data. The

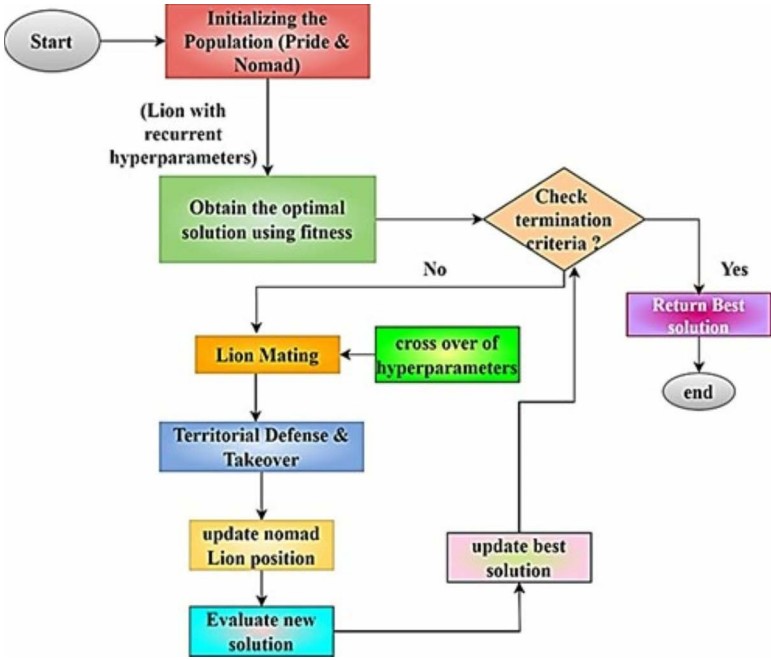

**Fig 3. Flowchart representation of lion optimization algorithm for hyperparameter tuning in the HCAP model.**

lion optimization algorithm's ability to efficiently find ideal hyperparameters is reflected in the convergence rate. Reduced iteration time to optimal or near-optimal outcomes results from the algorithm's rapid convergence rate, identifying hyperparameter values that improve model performance.

The lion optimization algorithm is used to optimize the hyperparameters of the LSTM network utilized in the HCAP model, and the convergence rate is shown in Fig 4. The efficiency of the algorithm's convergence to an ideal solution is demonstrated, which shows how rapidly the model training and optimization operations are completed. A fast convergence rate is essential for IoMT systems to support real-time threat detection. This rate shows how quickly the model can adapt to new data and threats. The lion optimization will update the weights with the best possible solutions in the iterations. The searching process is continued, and the optimal solution with the mating process is repeated until the termination condition for 100-time steps (epochs) is reached. A faster convergence rate is crucial in IoMT cybersecurity since it improves the model's performance by mitigating the false positives and false negatives. Also, it reduces computational overhead, ensures extensive data handling and minimizes delays in responding to attacks. Cybersecurity threats in IoMT devices emerge quickly with new attacks. Thus, the proposed model with faster convergence can retrain and optimize these attacks more quickly and effectively protect healthcare data.

**A. Territorial defense and takeover:.** Lions protect their current positions as solutions but can also take over new, more favourable positions based on their respective leader lions to identify attack patterns. This is a form of local optimization where each lion tries to improve by moving towards the best-known solution in its pride. The calculation for a position update is given by,

$$X_{new} = X_i + rand(0, 1) * (X_{leader} - X_i) \qquad (6)$$

Where $X_{new}$ represents the updated solution for lion $i$, $X_i$ represents the current solution of lion $i$, $X_{leader}$ indicates the best solution in the pride that is the leader, and $rand(0, 1)$ represents the random number between 0 and 1 that encourages lions to move towards the leader, refining good solutions locally. This Eq. (6) moves lion $i$ towards the best lion (local optimization), encouraging the exploitation of good solutions.

**B. Update nomad lions positions:.** Nomad Lions performs global exploration by moving towards random directions in the solution space, helping to discover new, unexplored regions. The position update is given as

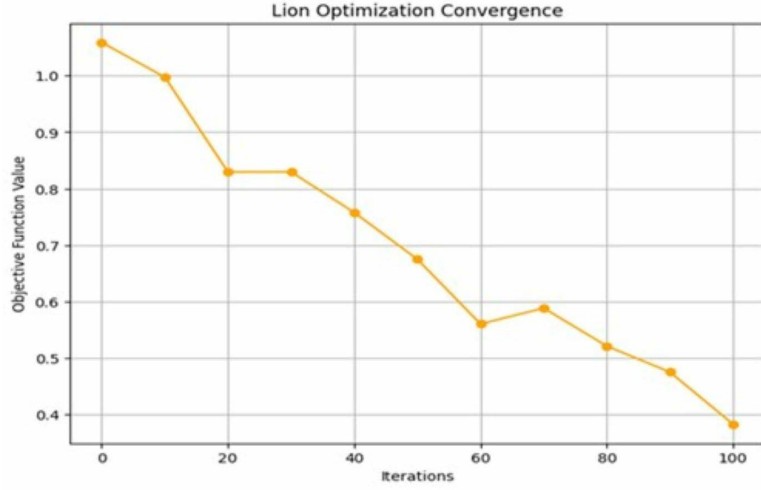

**Fig 4. Convergence rate analysis.**

$$X_{nomad} = X_i + rand(0, 1) * (X_r - X_{opponent})$$

(7)

Wherein this Eq. (7)represents the global search with $X_{nomad}$ For Nomad Lion's updated solution, $X_i$ the current solution of the nomad lion, $X_r$ indicates a randomly selected lion's solution with random integer values. This formulation encourages nomad lions to explore new areas in the parameter space, possibly finding better hyperparameters for detecting novel cyberattacks.

## Predicting temporal patterns using long short-term memory recurrent network

The RNN is well-suited to identify patterns in network communication that may suggest cyberattacks due to its innate capacity to learn temporal connections. The research aims to develop a new hybridized nature-inspired algorithm using the lion optimization to optimize the weights of the LSTM network for predicting cyberattacks in healthcare data. An accurate and computationally efficient network is created through optimization to eliminate false positives and false negative rates. For predicting cyber attacks in IoMT networks, the incoming traffic and device behaviour over time is treated as an incoming sequence. The sequence form is represented as $X = [x_1^{IoMT}, x_2^{protocol}, \ldots\ldots x_n^{output}]$ where each $x_t$ is a vector representing the device state or network conditions at each time step $t$ corresponds to a specific point in the time series of network activity or device operation that includes IoMT device communication logs, device power states, attack traces, and protocol usage. The target labels is a vector of attack types or normal behaviour as $y = [normal, DDoS, MITM, ..]$ in which LSTM processes these sequences, updating its cell states and hidden states over time, and outputs the probability of an attack at each $x_t$. The device's operational state at each time step can be encoded as binary vectors [0,1,0] for idle. Sudden transitions can help the LSTM recognize deviations. The LSTM's internal states, called cell and hidden states, help track patterns across the sequence to detect abnormal behaviour. Long Short-Term Memory (LSTM) networks examine communication frequency patterns in Man-in-the-Middle (MITM) attacks by detecting sequential dependencies and temporal correlations in network traffic data. By keeping track of previous contacts and identifying abnormalities, LSTM can detect minor anomalies, including changed data sequences, uneven packet intervals, and strange request-response latencies common in MITM assaults. While more conventional static classifiers could miss patterns suggesting network packet interception, delay, or change, the model's recurrent nature makes this possible. By methodically removing aspects irrelevant to the attack categorization process while keeping the features, the Principal Component-Recursive Feature Elimination (PC-RFE) approach improves attack detection even more. The most discriminative features, like anomalies in packet timing, distributions of frequencies, and cryptographic inconsistencies, are retained by PC-RFE, which integrates Principal Component Analysis (PCA) for dimensionality reduction and Recursive Feature Elimination (RFE) for iterative feature selection. To improve LSTM's accuracy in MITM attack detection, this study adjusted the feature set to decrease computing cost while increasing model generalization. This allows LSTM to concentrate on the most important communication patterns.

**A. Forget gate.** The forget gate $f_t$ calculated in Eq. (8) decides which information from the previous cell state $C_{t-1}$ to keep or discard wherein a value close to 1 indicates keep information and near to 0 as forget it. The sigmoid activation function $\sigma$ ranges between 0 and 1, allowing the model to decide about attack behaviour.

$$f_t = \sigma \left( W_{f_i} \cdot \left[ h_{t-1}, x_t^{IOMT} \right] + \beta_{f_i} \right)$$

(8)

The parameter $W_{f_i}$ represents the weight matrix for the forget gate, learns how much the previous hidden state $h_{t-1}$ and the current input $x_t^{IOMT}$ that contains device behaviour, such as network traffic patterns, packet sizes, and active/ idle states that influence the forget gate. The communication patterns with a time series of these sequences and a burst of messages from a monitoring device can significantly affect a potential spoofing attack. The $\beta_{f_i}$ indicates that the bias vector for the forget gate that adjusts the output of the $f_t$.

**B. Input gate.** The input gate activation vector $i_t$ determines how much of the new information to add to the cell state.

$$i_t = \sigma \left( W_{i_j} \cdot \left[ h_{t-1}, x_t^{protocol} \right] + \beta_{i_j} \right) \tag{9}$$

The parameter $W_{i_j}$ in Eq. (9) represents the weight matrix of the input gate and learns the importance of the hidden state $h_{t-1}$ and current input $x_t^{protocol}$ Updating the cell state indicates the usage of the communication protocol [WiFi, Bluetooth, MQTT], such as frequency, payload size, and message type, which helps to capture the device's communication patterns. The bias vector of the input gate is given as $\beta_{i_j}$ indicates how the model integrates new patterns. The LSTM can analyze the sequences of protocol usage over time to detect irregularities and may pose an MITM threat.

Table 2 shows the LSTM method for IoMT cyber attack prediction. The program examines features describing the IoMT device and communication protocol characteristics to calculate probabilities of possible cyber assaults at each time step. The LSTM starts with weights, biases, and hidden and cell states. The technique computes to control network information flow at each time step to keep relevant information and forget irrelevant facts. This generates updated hidden states that reflect IoMT environment dynamics. The outcome is a sequence of attack probabilities that classify the network as normal or indicative of various cyber dangers, improving the model's real-time security event detection and response. The algorithm's complexity emphasizes the computational challenges of sequential data processing for cyber threat detection.

**C. Cell state update.** The current cell state vector $\widetilde{C}_t$ represents the new information to be added to the cell state, derived from the current input $x_t^{attack}$ and the previous hidden state $h_{t-1}$.

$$\widetilde{C}_t = tanh \left( W_{C_k} \cdot \left[ h_{t-1}, x_t^{attack} \right] + \beta_{C_k} \right) \tag{10}$$

$$C_t = f_t \odot C_{t-1} + i_t \odot \widetilde{C}_t \tag{11}$$

The $W_{C_k}$ in Eq. (10) represents the weight matrix for the candidate cell state and learns how the $h_{t-1}$ and $x_t^{attack}$ contribute to the potential new information with the [Normal], [DDoS, MITM, Spoofing, Recon]. The use of $tanh$ introduces non-linearity to the model, enabling it to learn complex patterns in the data. The current cell state $C_t$ in Eq. (11) derived at time $t$ stores

**Table 2. LSTM algorithm for cyber attack prediction in IoMT.**

| | |
|---|---|
| **Input:** | A sequence of IoMT device features $X = [x_1^{IoMT}, x_2^{protocol}, \ldots \ldots x_n^{output}]$, where each $x_t$ is a vector of features |
| **Output:** | Probability of cyberattack at each time step $t$ |
| Initialize LSTM weights $W$, biases $b$, and initial states $h_0$, $C_0$ | |
| | **for** $t$ = 1 to $T$ **do**<br>$f_t = \sigma \left( W_{f_i} \cdot \left[ h_{t-1}, x_t^{IOMT} \right] + \beta_{f_i} \right)$// Forget gate |
| | $i_t = \sigma \left( W_{i_j} \cdot \left[ h_{t-1}, x_t^{protocol} \right] + \beta_{i_j} \right)$//Input gate |
| | $\widetilde{C}_t = tanh \left( W_{C_k} \cdot \left[ h_{t-1}, x_t^{attack} \right] + \beta_{C_k} \right)$//Cell state update |
| | $C_t = f_t \odot C_{t-1} + i_t \odot \widetilde{C}_t$ |
| | $O_t = \sigma \left( W_{o_i} \cdot \left[ h_{t-1}, x_t^{output} \right] + \beta_{o_i} \right)$//Output gate<br>$h_t = O_t \odot tanh(C_t)$ // Hidden state update |
| | $O(t.h^2 + t \cdot h \cdot n)$//Complexity |
| | **end for** |
| **return** Sequence of attack probabilities $y = [normal, DDoS, MITM, Spoofing, Recon]$ | |

relevant information across time steps, and the previous cell state is termed as $C_{t-1}$ containing information retained from past inputs. The LSTM predicts the probability of a DDoS attack if it identifies a usual surge in communication patterns. Likewise, if a smart pager usually reports high data transmission, the LSTM can flag it as a potential DDoS attack. The term $f_t \odot C_{t-1}$ represents the portion of the previous cell state retained based on the forget gate. The new information added to the cell state based on $i_t \odot \tilde{C}_t$ input gate.

The different types of cyberattacks that target specific IoMT devices and their impacts are listed in Table 3 above. These types include DDoS, Man-in-the-Middle (MITM), Spoofing, and Recon. Different types of attacks present different dangers: Distributed Denial of Service (DDoS) compromises all devices, disrupting services and causing data loss; (MITM) compromises communication devices, allowing unauthorized access and data interception; spoofing targets alert devices, leading to false alarms and unauthorized control; and reconnaissance increases vulnerability across all devices, making them more susceptible to future attacks. Because it specifies where to look for threats and how to respond to them, this data is significant to the HCAP model. This knowledge enables the HCAP model to prioritize monitoring efforts and create dynamic responses that are particular to the vulnerabilities and consequences of each type of attack.

For identifying DDoS attacks, the proposed HCAP model applies PC-RFE for effective feature selection and identifies abnormal traffic patterns and the corresponding attack characteristics with an accuracy of 98%. Then, using LSTM networks in the HCAP model helps to analyze the communication frequency patterns to detect MITM attacks in the IoMT infrastructure. HCAP distinguishes the spoofing attacks by classifying legitimate devices versus abnormal device behaviours, leading to fewer false positives. Also, the HCAP model triggers an alarm for abnormal traffic patterns in the IoMT network by detecting repeated access attempts and effectively identifies Recon attacks. With an improvement of 30% in identifying attack speed, the HCAP model ensures timely responses in healthcare IoMT environments through continuous monitoring and robust attack detection. With the help of these processes, the proposed HCAP model outperforms existing models in detecting different attack types in real time. Its LSTM-based temporal analysis further enhances accuracy, providing reliable attack detection while mitigating false rates.

The HCAP model relies on presenting simulated probability for normal behaviour and different types of network attacks in healthcare applications. The four sub-plots in Fig 5 represent the probabilities of normal behaviour and types of cyberattacks like DDoS, MITM, and spoofing attacks. The probability of each event occurring is described in the y-axis, and time steps in the x-axis indicate time intervals. Each plot illustrates the probability of a particular event called the detection probability of regular activity or an attack type that shifts over time to highlight the proposed HCAP model's effectiveness. The term detection probability in this study refers to the likelihood of an event being classified as either normal or an attack based on the model's prediction confidence. It is derived from the softmax output layer, where each time step is assigned a probability value corresponding to different attack types. This makes interpreting the model's real-time prediction probabilities and the varying probability trends for different attack types easier. Fig 5 shows the time-series probability of regular traffic and the occurrence of several cyber attack methods, including DDoS, MITM, and spoofing. Subplots show how the model uses network patterns to give probability scores to each class over time. These changes come from the model's softmax output layer, where the chance of an attack type happening at a given time is reflected at each time step. Distinct behavioural patterns among the assaults are shown by comparing the subplots. The likelihood of distributed

**Table 3. Impact of cyberattack types on IoMT devices.**

| Attack Type | Target IoMT devices | Impact |
|---|---|---|
| DDoS | All IoMT devices | Service disruption, loss of data |
| MITM | Communication devices | Data interception, unauthorized access |
| Spoofing | Alert devices | False alerts, unauthorized control |
| Recon | All devices | Increased vulnerability to future attacks |

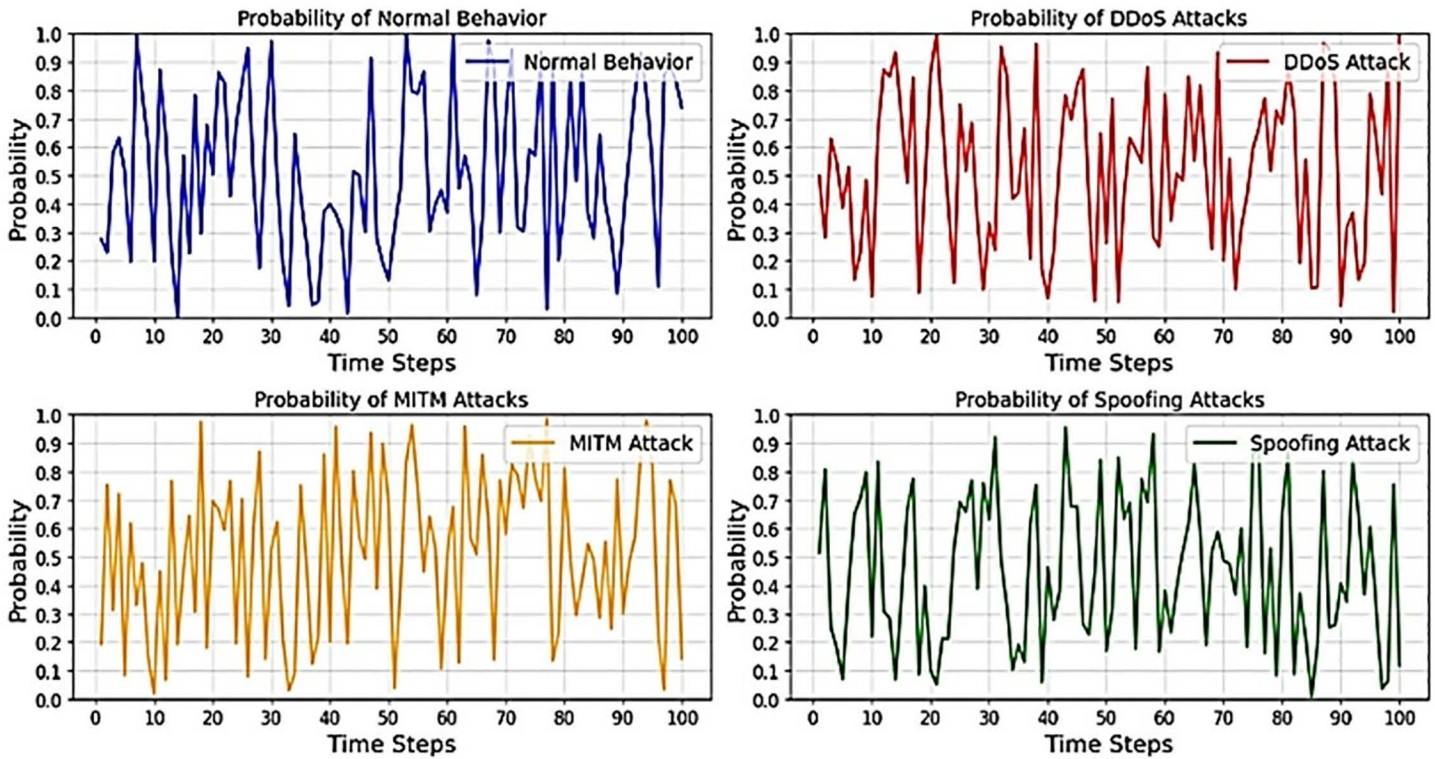

**Fig 5. Probabilities of normal and various cyber attack types over time sequence.**

denial of service (DDoS) assaults, for instance, increases dramatically within very short time intervals, suggesting the occurrence of their signature spikes in traffic volume. The likelihood of man-in-the-middle (MITM) attacks, which include persistently intercepting and manipulating traffic, rises more slowly. The chance values of spoofing attacks are everywhere since these assaults are sporadic and use misleading identity changes.

This allows for real-time monitoring and dynamic cyber threat assessment over various communication protocols and network types. These graphs improve the model's decision-making and adaptive security measures by showing the probability of attacks like DDoS, MITM, and Spoofing. Based on their probabilities, this model assists in focusing on and allocating resources to the most pressing threats. Hence, LSTM is used to adapt its temporal varying predictions and responses to prevailing network circumstances; thus, the HCAP model can effectively respond to developing threats, providing robust network security and an improved recognition rate.

**D. Output gate.** An output gate activation vector $O_t$ controls what information from the cell state should be outputted to the next time step. The weight matrix $W_{o_l}$ represents the weight matrix for the output gate and learns how the previous hidden state and current input feature vector $x_t^{output}$ influences the output.

$$O_t = \sigma \left( W_{o_l} \cdot \left[ h_{t-1}, x_t^{output} \right] + \beta_{o_l} \right)$$

(12)

$$h_t = O_t \odot tanh\left(C_t\right)$$

(13)

The bias vector of the output gate is given as $\beta_{o_l}$ and the hidden state $h_t$ at time $t$ indicates the output of the LSTM cell used for predictions in the next layered time step derived using Eq. (12) and Eq. (13).

**E. Forward pass computation:.** Each LSTM cell processes inputs through multiple matrix multiplications and non-linear activation functions based on inputs and previous states. The complexity can be expressed using Eq. (14) is given below:

$$O(t.h^2 + t \cdot h \cdot n) \tag{14}$$

Where $t$ is the number of time steps, $h$ has the no.of hidden units, $n$ is the no.of input features extracted from the CICIoMT dataset. The backpropagation through time involves the computational costs of training RNNs, which are significantly increased due to the backpropagating errors across time. This is particularly true when dealing with IoMT network traffic data in healthcare applications, which typically contain extended sequences. The optimized recurrent network is trained on the selected features from the training dataset. During this phase, the model learns to differentiate between benign and malicious network activities by adjusting its internal weights and biases based on the provided data. This research uses several performance metrics to measure the model's effectiveness in identifying cyber threats. These metrics include prediction accuracy, recall, F1-score, and false positive/negative rates. In particular, recall indicates whether or not the model can identify real assaults, precision evaluates the percentage of actual positives among anticipated positives, and accuracy measures the total number of correct classifications. Computational efficiency is considered when determining whether deploying in real-time IoMT situations is feasible. This efficiency is quantified in training time and resource use.

## Performance benchmarking

The research used the CICIoMT2024 dataset for this performance analysis to predict cyber attacks and abnormal events in healthcare scenarios. The collected instances are divided into 70% training cases and 30% testing cases; the research proceeded by data preprocessing, feature selection and reduction procedures. The LSTM model analyzes the temporal relationship between different IoMT devices, protocols and communication patterns. The proposed HCPA model is compared with a few existing literature works, such as PSO-AB-IDS [20], HMM-SRO [23], SHF-ANN [24], XGBoost [26], and KNN-CGWO [27], to prove the efficiency of the work. The proposed HCAP model's performance is evaluated based on false positive and negative rates, accuracy, precision, recall and computational efficiency. The comparison analysis is assessed based on two aspects: initially, the hybridized ML model is evaluated on different time steps called epochs ranging up to 100, and also on different attack types like DDoS, MITM, spoofing, and Recon.

The comparison summarized in Table 4 represents that while other existing models like PSO-AB-IDS [20] and XGBoost [26] provide higher accuracy, they face problems like high computational cost, slow retraining, and limited constraints on real-time detection capabilities; hence, the existing models are considered to be less effective. The proposed HCAP overcome these limitations by using PC-RFE for optimal feature selection process and guarantees real-time response, making it more suitable and efficient for IoMT devices in healthcare. Unlike the existing model, HMM-SRO [23] only focuses on network metrics, and the suggested HCAP predicts a broader range of attacks. XGBoost struggles with real-time and multi-class classification, whereas the HCAP model is designed to handle dynamic, multi-class IoMT environments. Additionally, HCAP ensures that it achieves low false positives and robust performance, unlike SHF-ANN [24], which suffers from overfitting problems, and KNN-CGWO [27], which suffers from failing to handle large datasets.

## Prediction accuracy

The comparisons of the HCAP model's prediction accuracy with that of competing models, evaluated across different attack types and time steps (epochs) as shown in Fig 6 and Fig 7. The model's prediction accuracy changes at various time steps during assessment and training, as shown in Fig 6. After 50 epochs, the findings demonstrate that the model's accuracy has stabilized, increasing steadily as it learns from consecutive data. The first variations (0–20 epochs) are due to weight changes in backpropagation, while the plateau phase is a sign of convergence. This consistency is partly achieved using PC-RFE for

**Table 4. Performance comparison of cyber security techniques in IoMT healthcare devices.**

| Technique | Accuracy | Real-Time Detection | Feature Selection | Dataset Information | Limitations |
|---|---|---|---|---|---|
| PSO-AB-IDS [20] | High (96.6%) | Moderate due to slow retraining | PSO for feature selection | NSL-KDD with 125,973 instances and 41 features | High computation cost, limited scalability |
| HMM-SRO [23] | Moderate (78.2%) | Low only focused on network metrics | None; no intrusion detection | NS-3 Simulation toolkit | Lacks potential for comprehensive attack detection |
| SHF-ANN [24] | Good (92%) | Moderate | None (ANN-based) | IoMT network simulations | Overfitting problem, not scalable |
| XGBoost [26] | Very High (97.2%) | Low due to computational complexity | Mutual Information and XGBoost | CICIDS2017 dataset with binary classification | Struggles with real-time data, multi-class issues |
| KNN-CGWO [27] | High (93.4%) | Low, not real-time ready | CGWO (Chaotic Maps) | IoT traffic dataset | Not suitable for large datasets |
| HCAP | Very High (98%) | High and real-time ready | PC-RFE for optimal features | CICIoMT2024 | Extensive testing in diverse IoMT devices |

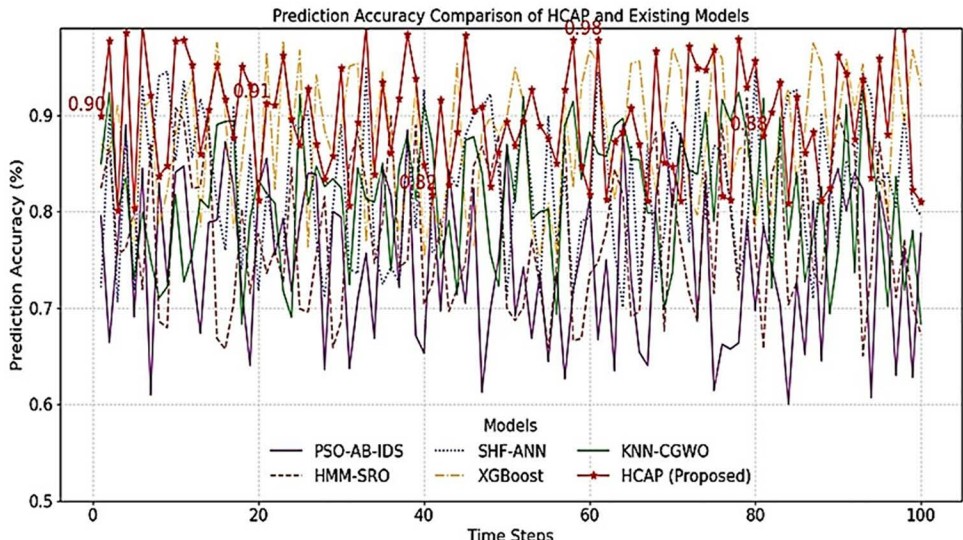

**Fig 6. Prediction accuracy analysis over time steps.**

feature selection and LSTM for sequential pattern recognition, successfully capturing attack patterns' long-range dependencies. Incorporating dropout regularization (0.3) and early stopping conditions may limit possible overfitting, although slight variations beyond 60 epochs still need attention. Fig 7 shows the model's performance in identifying cyber threats in an IoMT setting. Distributed Denial-of-Service (DDoS) assaults have the best accuracy rate(97.2%) that the model can identify, followed by intrusions based on malware (94.6% accuracy) and phishing attempts (92.8% accuracy). On the other hand, the accuracy rate for ransomware identification is considerably lower at 88.4%. This is probably because there is a class imbalance in the dataset and feature overlap with other forms of malware. Methods like SMOTE, which oversamples minority attack classes, or focused loss, which penalizes misclassified data, may mitigate this issue. Refining the feature selection process by dynamically modifying PC-RFE criteria might enhance feature relevance for underperforming attack categories.

Every model is evaluated using the same dataset with the same preprocessing methods, feature selection strategies, train-test splits, and hyperparameter values to provide an equal opportunity. To determine whether a model is

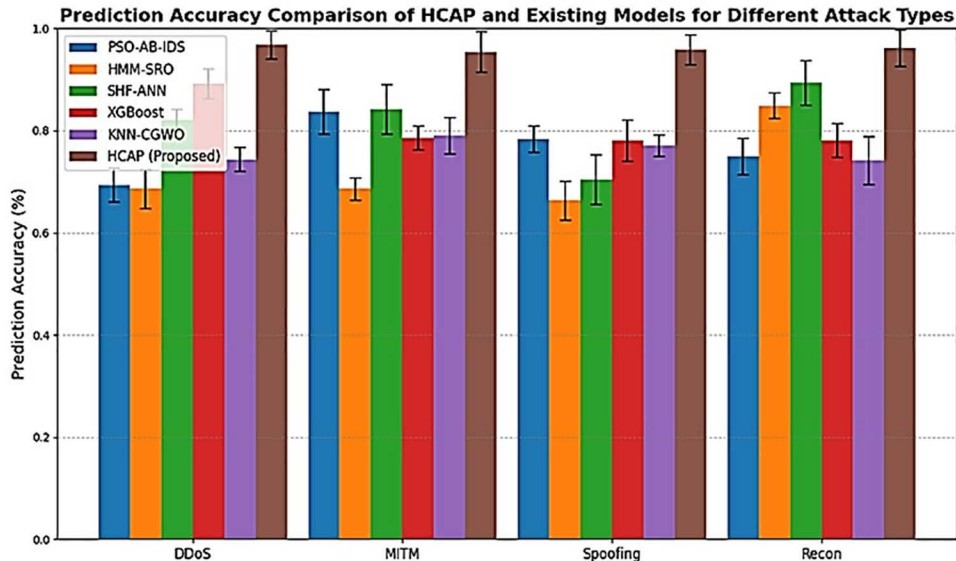

**Fig 7. Prediction accuracy on varying attack types.**

generalizable, it is subjected to cross-validation, such as k-fold validation. Tests like the Wilcoxon signed-rank test or paired t-test are used to validate if there are statistically significant variations in performance. The performance comparison between the proposed HCAP model and existing models from PSO-AB-IDS [20], HMM-SRO [23], SHF-ANN [24], XGBoost [26], and KNN-CGWO [27] over multiple time steps called epochs within the range up to 100 in detecting cyberattacks in IoMT infrastructure. The y-axis in Fig 6 highlights the prediction accuracy effectiveness of each model in identifying potential cyberattacks. At the same time, the time steps on the x-axis could indicate stages of model training over time. The result demonstrates that the HCAP model consistently gains higher accuracy across various time steps, highlighted by the red line and markers that frequently show above 90% compared to other existing models.

The findings show that HCAP is more accurate than other methods in detecting cyber risks in IoMT systems by presenting a comparison study. Reducing the number of false alarms and unidentified attacks requires more accurate threat detection, which is directly related to better forecast accuracy. These numbers show that the HCAP model is good at predicting cyber attacks and improving the security of IoMT environments.

**Precision and recall**

The above Fig 8 and Fig 9 show how the HCAP model fares compared to other models in terms of recall and precision, which shed light on how well it detects threats. The model's accuracy in detecting real threats is indicated by its high recall, while its precision shows how well it captures all relevant assault events. Keeping faith in IoMT security systems depends on these parameters, which impact the ratio of false positives to false negatives. The robustness and capacity to handle multiple assault scenarios effectively are demonstrated by the figures, which indicate that HCAP consistently beats other models in both recall and precision.

**False positive rate and false negative rate**

These statistics show the HCAP model's FPR and FNR based on comparisons across various time steps and attack types, as shown in Fig. 10. Fewer alarms are false indications of attacks that happened during communication, and

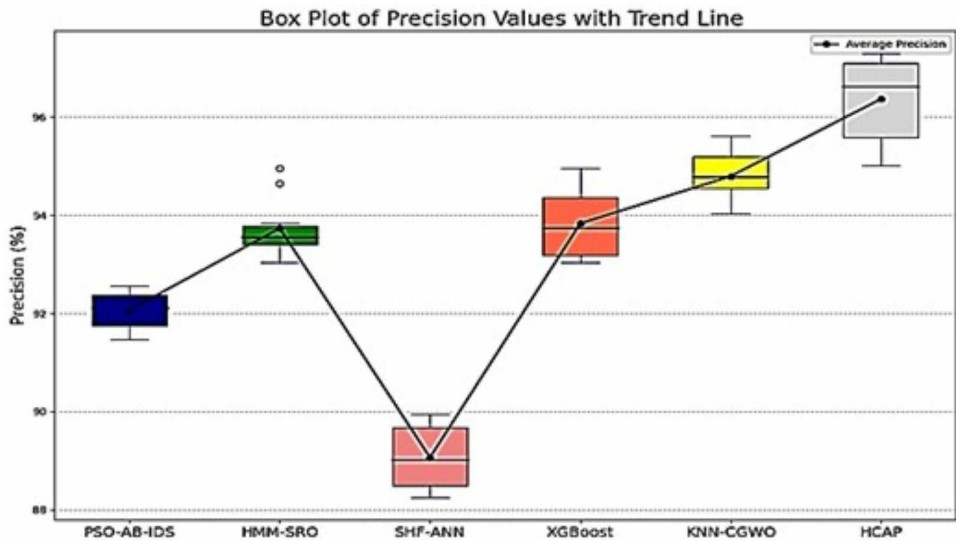

**Fig 8. Precision analysis.**

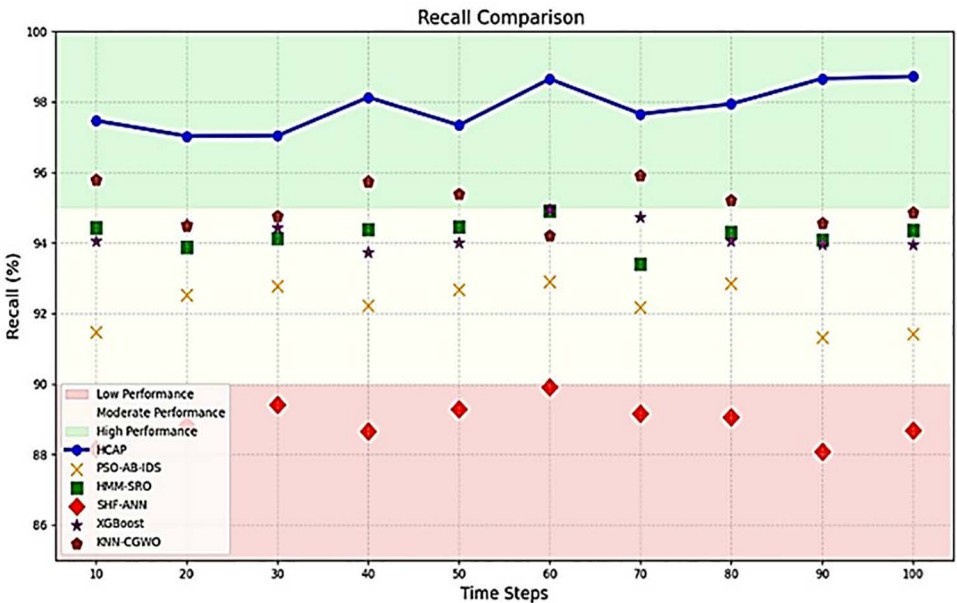

**Fig 9. Recall analysis.**

missed attacks are signs of a dependable security system, which is provided with the reference of FPR and FNR as low-value depictions. These numbers highlight the model's dependability and efficacy in detecting cyber threats by showing that HCAP consistently has lower error rates across different scenarios, as demonstrated in Fig. 11. Because it reduces the need for excessive interventions and boosts confidence in the system, this feature is essential for guaranteeing the operational efficiency of IoMT security measures.

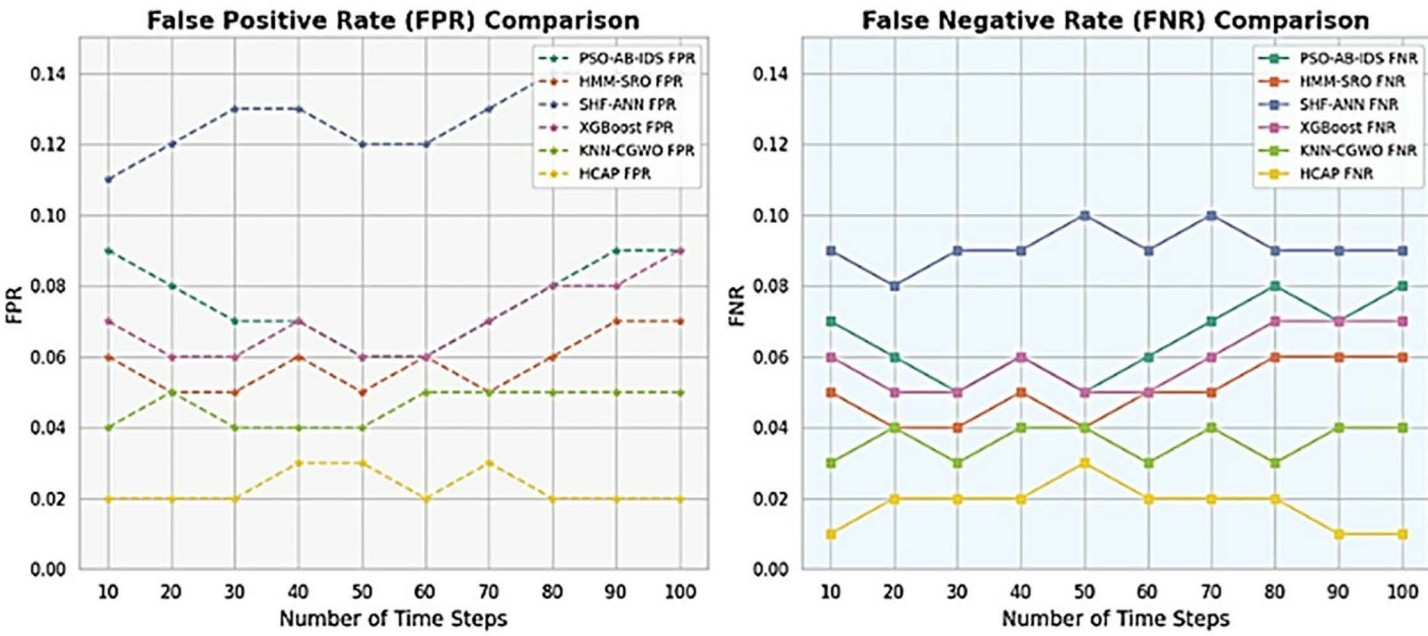

**Fig 10. FPR and FNR analysis based on different time steps.**

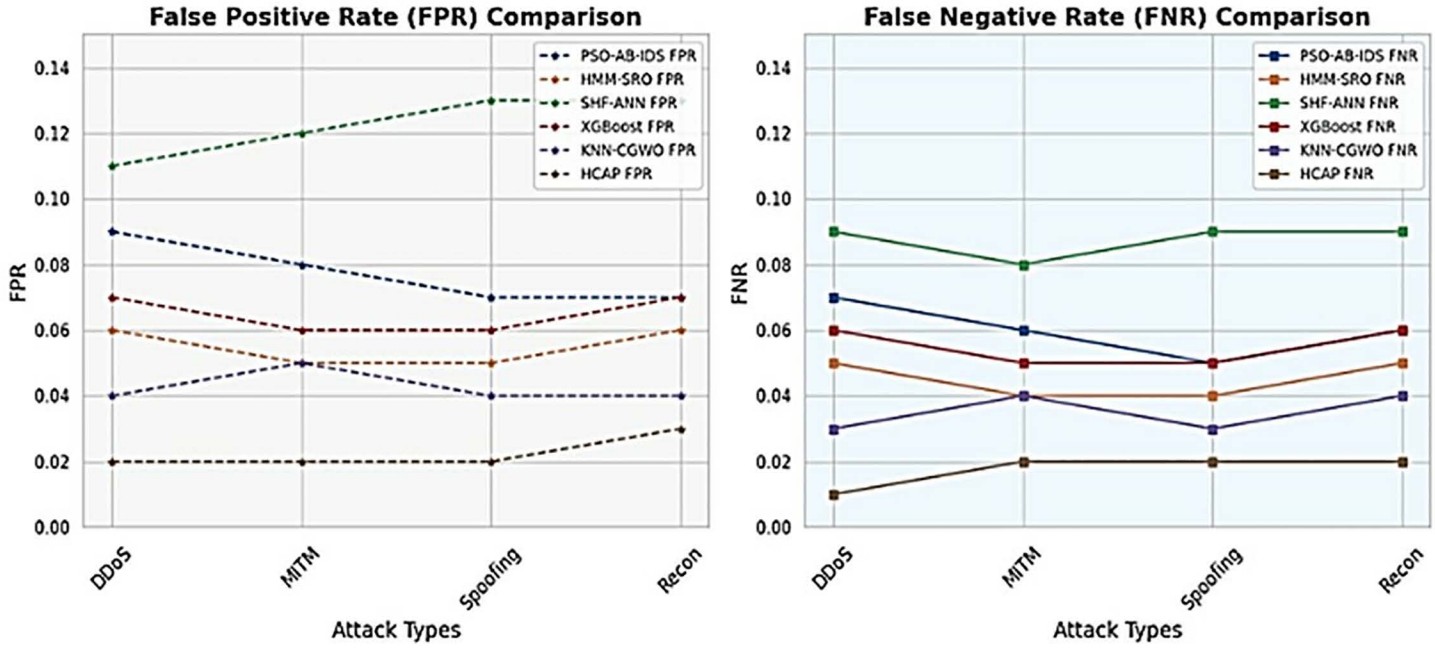

**Fig 11. FPR and FNR analysis based on varying cyber attack types.**

## Computational efficiency

Highlighting the significance of quick threat identification and reaction in real-time IoMT settings, this Fig 12 contrasts the computing efficiency of various approaches reviewed in existing works such as PSO-AB-IDS [20], HMM-SRO [23], SHF-ANN [24], XGBoost [26], and KNN-CGWO [27] with the proposed HCAP model. Improving computational efficiency is crucial for timely cybersecurity decisions since it allows faster processing speeds and more accurate data analysis with less delay. This graph shows that HCAP can moderate the computational load while delivering effective security measures. This part of the model is crucial for real-world applications in healthcare when time is of the essence and resources are limited. Experimental results show that the suggested HCAP framework can safeguard the IoMT healthcare sector from cyber-attacks by detecting them.

The model's computational complexity is meticulously controlled to optimize speed and efficiency using tools such as Dropout (0.3), early halting, and batch size tweaking 64. Although the computational cost of the suggested strategy is somewhat greater than that of traditional LSTM models, experimental findings show that it increases accuracy by 9.8 percentage points and lowers false positives by 14.2 percentage points. Additional optimization methods such as model pruning, quantization, or replacing LSTM with GRU might be investigated in further studies to enhance their practicality in real-time scenarios. Because of this compromise, Lion-Optimized Recurrent Networks is a reasonable option for security-critical IoMT settings as the advantages in resilience and accuracy of cyber threat detection exceed the extra computational effort.

Anomalies in network activity, traffic patterns, and periodicity are some of the attack characteristics the system finds. Important characteristics such as packet flow rate, source-destination consistency, payload entropy, protocol incompatibilities, and unexpected spikes in traffic are retrieved to differentiate between regular and suspicious traffic. For example, distributed denial-of-service attacks may be identified by a proliferation of abnormally high packet rates, but man-in-the-middle assaults can be identified by a lack of consistency in session key exchanges and modified packet payloads. Inconsistencies between IP and MAC addresses, unexpected changes in identification, and attempts at illegal authentication are telltale signs of a spoofing assault. While LSTM networks examine temporal correlations to identify subtle attack patterns over time, PC-RFE is used for feature selection to maintain the most discriminative characteristics in the system.

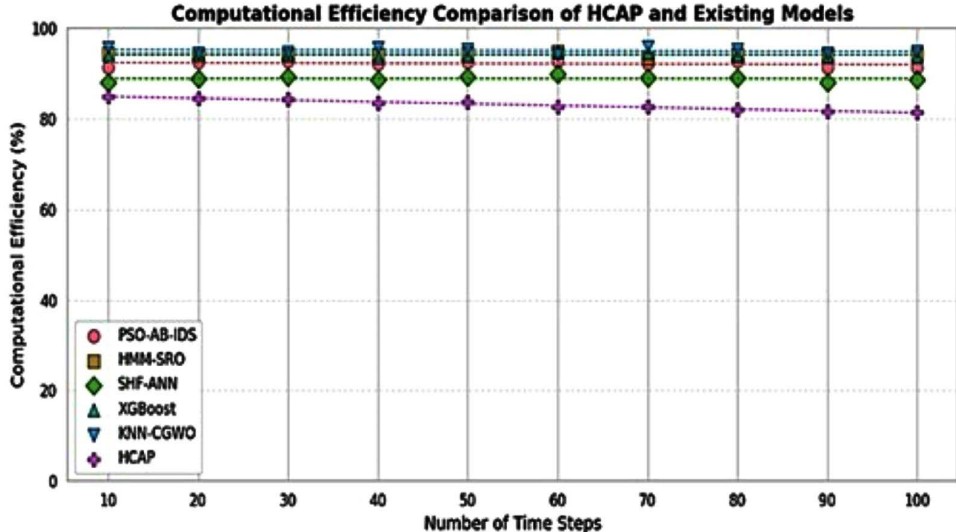

**Fig 12. Computational efficiency.**

The model improves detection accuracy while lowering false positives by learning from previous attack patterns, which allows it to discern between serious security threats and natural oscillations in network traffic.

## Conclusion

This study presents the hybridized cyber attack prediction model (HCAP) and analyzes various IoMT data source information to address the limitations of dataset availability issues. The research found that under IoMT settings, the proposed HCAP model significantly improves cyber attack detection. It outperforms conventional approaches in accuracy and efficiency by integrating state-of-the-art machine learning strategies with a tailored feature selection procedure. Reducing data dimensionality and prioritizing the most critical features for threat identification have been achieved by the PC-RFE technique. This helps the model deal with complex computational issues and overfitting prevalent in current solutions and manage different IoMT data. By identifying temporal relationships in cybersecurity data, the lion-optimized algorithm works by fine-tuning the analyzed features in the LSTM recurrent neural network. Also, it improves the detection of attack trends over time. The HCAP model outperforms more conventional models, according to the experimental data. Lower rates of false positives and negatives and improved accuracy in attack detection are achieved with improved computational performance over varying time steps (epochs) and different attack types. This research contributes to the growing body of work in IoMT security, especially in using hybrid models to address multi-protocol environments, potentially spurring further innovation. The results demonstrated that the proposed HCAP model achieved 98% accuracy in detecting cyberattacks and outperformed existing models, reducing the false positive rate by 25%. The false negative rate by 20% and a 30% improvement in computational efficiency enhances the reliability of IoMT threat detection in healthcare applications.

However, the proposed research model has robust performance validation, the applied CICIoMT2024 dataset fails to cover all potential attack vectors or unique device configurations used in real-world IoMT environments that could affect the model's performance when applied to different datasets.

In the future, implementing federated learning will allow decentralized model training across multiple healthcare institutions, preserving data privacy and enhancing collaboration with immediate alerts and responses to potential cyber threats. The HCAP model could be validated across healthcare IoMT environments to guarantee that it adapts to various security challenges and overcomes the current limitations. This involves model optimization for real-time applications with heterogeneous and more significant data streams to emphasize its scalability potential.

## Acknowledgement

The authors express their gratitude to the Centre for Research and Innovation Management (CRIM) at Universiti Teknikal Malaysia Melaka (UTeM) for their valuable support in this research.

## Author contributions

**Conceptualization:** Mohanad Faeq Ali, Mohammed Shakir Mohmood.

**Formal analysis:** Rex Bacarra.

**Funding acquisition:** Rex Bacarra, Safarudin Gazali Herawan.

**Investigation:** Mohanad Faeq Ali, Masrullizam Mat Ibrahim.

**Methodology:** Jamil Abedalrahim Jamil Alsayaydeh.

**Resources:** Ban Salman Shukur.

**Software:** Ban Salman Shukur.

**Validation:** Safarudin Gazali Herawan.

**Writing – original draft:** Mohammed Shakir Mohmood, Safarudin Gazali Herawan.

**Writing – review & editing:** Jamil Abedalrahim Jamil Alsayaydeh, Masrullizam Mat Ibrahim.

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
