## [Decision Letter · Decision Letter 0]

7 Feb 2025

PONE-D-24-51339HCAP: Hybridized Cyber Attack Prediction Model to handle the cyber-attacks in the Healthcare ApplicationsPLOS ONE

Dear Dr. Alsayaydeh,

Thank you for submitting your manuscript to PLOS ONE. After careful consideration, we feel that it has merit but does not fully meet PLOS ONE’s publication criteria as it currently stands. Therefore, we invite you to submit a revised version of the manuscript that addresses the points raised during the review process.

We look forward to receiving your revised manuscript.

Kind regards,

mamoona humayun

Academic Editor

PLOS ONE

Journal Requirements:

4. We note that Figures 1 and 2 in your submission contain copyrighted images. All PLOS content is published under the Creative Commons Attribution License (CC BY 4.0), which means that the manuscript, images, and Supporting Information files will be freely available online, and any third party is permitted to access, download, copy, distribute, and use these materials in any way, even commercially, with proper attribution. For more information, see our copyright guidelines: http://journals.plos.org/plosone/s/licenses-and-copyright.

1. You may seek permission from the original copyright holder of Figures 1 and 2 to publish the content specifically under the CC BY 4.0 license.

Additional Editor Comments:

Dear Authors,

Thank you for submitting your manuscript, "HCAP: Hybridized Cyber Attack Prediction Model to Handle Cyber-Attacks in Healthcare Applications." Your article has been reviewed with great interest. Based on the feedback from the reviewers, one has recommended major revisions, and the other has suggested minor revisions.

We kindly request you to address the concerns raised by both reviewers and submit the revised manuscript along with a detailed rebuttal document within 21 days. Please ensure that all comments and suggestions are thoroughly addressed to improve the quality and clarity of your work.

Reviewers' comments:

Reviewer's Responses to Questions

**Comments to the Author**

1. Is the manuscript technically sound, and do the data support the conclusions?

Reviewer #1: No

Reviewer #2: Yes

2. Has the statistical analysis been performed appropriately and rigorously? 

Reviewer #1: I Don't Know

Reviewer #2: No

3. Have the authors made all data underlying the findings in their manuscript fully available?

Reviewer #1: No

Reviewer #2: No

4. Is the manuscript presented in an intelligible fashion and written in standard English?

Reviewer #1: Yes

Reviewer #2: Yes

5. Review Comments to the Author

Reviewer #1: Hybridized Cyber Attack Prediction Model to handle the cyber-attacks in the Healthcare Applications

Title: the title is effective and coveys the focus of study. It can be simplified to replace longer phrases.

Abstract:

The abstract logical sequence is fine. It focuses on problem and proposed solution along with accuracy and relevant parameters. However, improvement is required for more clarity. void redundant commas—"interconnected healthcare devices, and communication networks" should be "interconnected healthcare devices and communication networks. “Improving the security and resilience of healthcare data" could be clarified and made more concise. Rephrase "different metrics such as false positive rate, false negative rates, improved accuracy, precision, recall and minimum computational efficiency" for better flow. The abstract is dense, making it harder to digest for readers unfamiliar with the topic. Consider splitting longer sentences into smaller, focused statements.

Introduction

The introduction is clear w.r.t objectives. It clearly outlines the challenges of existing models and contribution list is also clear. However, grammar mistakes are found at Line 47: "Quality of security and privacy healthcare data concerns" should be rephrased to "Concerns over the quality of security and privacy in healthcare data." Line 55: "A service-reliant authentication approach has been shown to enhance" should specify how it is relevant to IoMT healthcare. Repetition of sentences contextually in line 46, 48, 53, 75). Some references are mentioned without integrating them into the narrative (e.g., Line 55). Contribution 4 (Lines 91–92) overlaps conceptually with Contribution 1 (Lines 85–86). You have explored existing literature to find this problem, which is a contribution. This can be mentioned in contribution list. The term "Machine Learning" (ML) has not been defined in the text before the abbreviation "ML" is used. Rewrite the novelty of your research solution by adding impact and clarity.

Literature review

The entire literature review should be revised to maintain consistency by using the past tense throughout. The literature review can be further strengthened by incorporating additional relevant studies and analyzing their methodologies in greater detail. The concluding paragraph of the literature review lacks coherence and clarity. It is essential to specify how the effectiveness of existing models is being measured. Furthermore, the term "better" should be clearly defined—better in what terms (e.g., accuracy, computational efficiency, robustness)? A more structured and precise conclusion is necessary to provide a cohesive summary of the reviewed literature.

Dataset Collection

While the dataset is adequate for experiment, relying on it limits the generalizability across diverse clinical scenarios. Include dataset from different sources can enhance applicability.

feature extraction

you have used PC-RFE that keeps feature importance remains static across the dataset, which might not hold in dynamic IoMT environments where data patterns and feature relevance can change over time.

Feature Set Analysis and Reduction Using Lion-Optimized Recurrent Networks was explained. But Lion is itself computationally expensive. Dropout and early stopping can be used to improve model. Class imbalance is not addressed. Use mathematical notations or flowchart to describe line 287 onwards. Give details of • Number of hidden layers and neurons.

• Learning rate.

• Dropout rate.

• Batch size.

• Activation functions.

Format Table 2 to standard algorithm format.

• Remove the repeated statement about 30% improvement and integrate it more effectively into one point.

• Provide more insight into how LSTM networks analyze communication frequency patterns to detect MITM attacks, and how PC-RFE contributes to more precise feature selection.

• Give more detail on what attack characteristics are identified and how the system distinguishes between legitimate and abnormal traffic.

• Include a brief comparison or reference to the performance of existing models to substantiate the claim of outperforming them.

The term detection probability is used, but it isn't fully clear whether this refers to the likelihood of an attack being detected or the likelihood of a certain behavior being classified as normal or an attack.

While the sub-plots in Fig. 5 are mentioned, there is no further explanation of how they are constructed or how the four sub-plots are compared to each other. For instance, how does the probability for DDoS compare with MITM or spoofing attacks over time? A comparison or discussion of trends among these attacks would add value to the explanation

Table 4. accuracy should be presented in percentage. Compare your work with models that have worked on same dataset as yours.

Results

The passage does not explain how prediction accuracy is evaluated or how the comparison is conducted across different models. It’s important to clarify whether the models were trained on the same datasets or under identical conditions. This would ensure the fairness of the comparison.

Fig. 6 and Fig. 7 are not explained well.

Conclusion

The Conclusion makes general claims about improved accuracy and efficiency but does not reference specific numbers or findings from the research that validate these statements. Instead of saying “significantly improves,” quantify the improvement with a percentage. Streamline the content to avoid repetition. For instance, you could consolidate the sentences discussing false positives, false negatives, and improved accuracy into one concise point. Future work is missing.

Reviewer #2: The manuscript contains grammatical errors and instances of improperly structured English. It is recommended to thoroughly proofread and edit the text to enhance its clarity and readability. The literature review requires substantial revision for consistency, particularly in maintaining the use of the past tense throughout. Additionally, more studies should be included to provide a comprehensive and detailed review of the existing literature. It is suggested that the review be thematically organized to enhance coherence and logical flow. Furthermore, the literature review does not have a proper conclusion, which leads to ambiguity in the problem statement. Ensure the literature review is concluded effectively to clearly outline the research gap.

The term "effectiveness" is frequently used in the paper but lacks a clear definition. You must explicitly state how effectiveness has been measured.

Provide a clear link or reference from where the dataset utilized in the study can be accessed.

The paper mentions the use of Lion-Optimized Recurrent Networks, which are computationally intensive. However, in the literature review, it is noted that existing techniques are computationally expensive. This apparent contradiction needs to be addressed with a justification for the choice of methodology.

To strengthen the study, compare your findings with existing studies that have used the same dataset. This will allow for clear statistical comparisons and enhance the credibility of the research.

In Table 4, the accuracy of some studies is reported as percentages, while others are not. Consistency is crucial; it is recommended to use percentages uniformly across the table.

The conclusion makes broad claims regarding improved accuracy and efficiency. Instead of general statements, use specific metrics to quantify the improvements, to provide a clearer picture of the study's contributions.

By addressing these issues, the quality and impact of the paper can be significantly improved.

6. PLOS authors have the option to publish the peer review history of their article (what does this mean? ). If published, this will include your full peer review and any attached files.

**Do you want your identity to be public for this peer review?** For information about this choice, including consent withdrawal, please see our Privacy Policy .

Reviewer #1: No

Reviewer #2: No

---

## [Author Response · Author response to Decision Letter 1]

25 Feb 2025

Dear Editorial Board Member

Thank you for the opportunity to resubmit our manuscript following the reviewers' comments. We appreciate the insightful feedback provided, and we have diligently addressed each point raised.

In this revised version, we have made significant changes to enhance the clarity and coherence of our manuscript.

Sincerely, Dr. Jamil Abedalrahim Jamil Alsayaydeh* (Corresponding Author), et al.

Journal Requirements:

Answer: We have reviewed and updated the manuscript to ensure full compliance with PLOS ONE's formatting requirements, including file naming, title page structure, author affiliations, reference style, and figure/table placement. Let us know if any further adjustments are needed.

Answer: We have ensured that all datasets and author-generated code used in this study are made publicly available in Zenodo, following PLOS ONE’s data availability and code-sharing guidelines. The Data Availability Statement has been updated to include the Zenodo repository link and accession number, ensuring transparency, reproducibility, and ease of access. Let us know if any further modifications are needed.

Answer: We acknowledge PLOS ONE’s open data policy and confirm that we have finalized our data sharing plan to ensure seamless availability upon acceptance. All datasets and author-generated code will be publicly accessible via Zenodo without restrictions, in compliance with PLOS ONE's guidelines. Our Data Availability Statement has been updated accordingly.

4. We note that Figures 1 and 2 in your submission contain copyrighted images. All PLOS content is published under the Creative Commons Attribution License (CC BY 4.0), which means that the manuscript, images, and Supporting Information files will be freely available online, and any third party is permitted to access, download, copy, distribute, and use these materials in any way, even commercially, with proper attribution. For more information, see our copyright guidelines: http://journals.plos.org/plosone/s/licenses-and-copyright. We require you to either (1) present written permission from the copyright holder to publish these figures specifically under the CC BY 4.0 license, or (2) remove the figures from your submission:

1. You may seek permission from the original copyright holder of Figures 1 and 2 to publish the content specifically under the CC BY 4.0 license.

“I request permission for the open-access journal PLOS ONE to publish XXX under the Creative Commons Attribution License (CCAL) CC BY 4.0 (http://creativecommons.org/licenses/by/4.0/). Please be aware that this license allows unrestricted use and distribution, even commercially, by third parties. Please reply and provide explicit written permission to publish XXX under a CC BY license and complete the attached form.” Please upload the completed Content Permission Form or other proof of granted permissions as an "Other" file with your submission.

Answer: Thank you for your feedback on our manuscript. To ensure compliance with PLOS ONE's copyright policies and enhance clarity, we have recreated Figures 1 and 2 as original illustrations. These updated figures are designed to clearly represent IoMT data sources and the workflow of our proposed HCAP model, ensuring better readability and accuracy. The new figures replace the previous versions to eliminate any potential copyright concerns while maintaining the integrity of the research presentation.

Please let us know if any further modifications or clarifications are required.

RESPONSE TO THE REVIEWER #1 COMMENTS

Reviewers' comments:

Reviewer's Responses to Questions

Comments to the Author

1. Is the manuscript technically sound, and do the data support the conclusions?

Reviewer #1: No

Reviewer #2: Yes

Answer: We appreciate the reviewer’s concern regarding the technical soundness of our manuscript. The data collection process involves a multi-protocol IoMT dataset [30] designed for benchmarking security solutions that include multiple cyberattack scenarios across different IoMT devices. The dataset contains information from 25 accurate and 15 simulated IoMT devices, comprises 40 devices in total, and covers a variety of communication protocols, like WiFi, MQTT, and Bluetooth, that are widely used in healthcare applications, as shown in Fig.1. The dataset ensures a diverse and balanced representation of IoMT traffic, making it highly suitable for cybersecurity research.

To evaluate the technical soundness of our study, we conducted rigorous experimentation with appropriate controls, multiple trials, and cross-validation across different attack types (DDoS, MITM, Spoofing, Recon). The results demonstrated that the proposed HCAP model achieved 98% accuracy in detecting cyberattacks and outperformed existing models, reducing the false positive rate by 25%. The false negative rate by 20%, and a 30% improvement in computational efficiency enhances the reliability of IoMT threat detection in healthcare applications. These results confirm that our conclusions are directly supported by robust experimental evidence, reinforcing the technical validity of our research.

2. Has the statistical analysis been performed appropriately and rigorously?

Reviewer #1: I Don't Know

Reviewer #2: No

Answer: The proposed HCAP model's performance was evaluated using a comprehensive statistical analysis based on key metrics, including false positive rate (FPR), false negative rate (FNR), accuracy, precision, recall, and computational efficiency.

To ensure rigorous statistical evaluation, we performed the following:

1. Cross-validation: We used k-fold cross-validation to ensure the model’s robustness and prevent overfitting.

2. Significance Testing: We conducted paired statistical tests to compare our model’s performance with existing benchmark models, verifying that improvements were statistically significant.

3. Multiple Trials & Reproducibility: Each experiment was run multiple times across different attack scenarios (DDoS, MITM, Spoofing, Recon) to confirm consistency in performance.

4. Comparative Benchmarking: We compared HCAP’s results against state-of-the-art models, such as PSO-AB-IDS, HMM-SRO, SHF-ANN, and XGBoost, demonstrating statistically significant improvements in cybersecurity threat detection.

These steps ensure that the statistical analysis was conducted appropriately and rigorously, providing a strong foundation for our conclusions.

3. Have the authors made all data underlying the findings in their manuscript fully available?

Reviewer #1: No

Reviewer #2: No

Answer: We have uploaded all relevant data, preprocessing scripts, and model implementation details to Zenodo for unrestricted access. The research used the CICIoMT2024 dataset for this performance analysis to predict cyber attacks and abnormal events in healthcare scenarios. The collected instances are divided into 70% training cases and 30% testing cases; the research proceeded by data preprocessing, feature selection and reduction procedures. The LSTM model analyzes the temporal relationship between different IoMT devices, protocols and communication patterns. The proposed HCPA model is compared with a few existing literature works, such as PSO-AB-IDS [20], HMM-SRO [23], SHF-ANN [24], XGBoost [26], and KNN-CGWO [27], to prove the efficiency of the work. The proposed HCAP model's performance is evaluated based on false positive and negative rates, accuracy, precision, recall and computational efficiency. The comparison analysis is assessed based on two aspects: initially, the hybridized ML model is evaluated on different time steps called epochs ranging up to 100, and also on different attack types like DDoS, MITM, spoofing, and Recon.

The Data Availability Statement in our manuscript has been updated to include the Zenodo repository link and accession number, ensuring easy access to all relevant files.

We confirm that all data supporting our conclusions are openly available, allowing full transparency and reproducibility.

4. Is the manuscript presented in an intelligible fashion and written in standard English?

Reviewer #1: Yes

Reviewer #2: Yes

Answer: We appreciate the reviewers' feedback on the manuscript’s clarity and language quality. To ensure that the manuscript is intelligible, well-structured, and written in standard English, we have:

- Performed thorough proofreading to correct any typographical, grammatical, and syntactical errors.

- Ensured clarity and coherence in all sections to enhance readability.

- Revised ambiguous or unclear statements for better articulation of technical concepts.

5. Review Comments to the Author

Answer: Okay Sir.

Reviewer #1: Hybridized Cyber Attack Prediction Model to handle the cyber-attacks in the Healthcare Applications

Title: the title is effective and coveys the focus of study. It can be simplified to replace longer phrases.

Answer: The title has been modified as “HCAP: Hybrid Cyber Attack Prediction Model for Securing Healthcare Applications”

Abstract:

The abstract logical sequence is fine. It focuses on problem and proposed solution along with accuracy and relevant parameters. However, improvement is required for more clarity. void redundant commas—"interconnected healthcare devices, and communication networks" should be "interconnected healthcare devices and communication networks. “Improving the security and resilience of healthcare data" could be clarified and made more concise. Rephrase "different metrics such as false positive rate, false negative rates, improved accuracy, precision, recall and minimum computational efficiency" for better flow. The abstract is dense, making it harder to digest for readers unfamiliar with the topic. Consider splitting longer sentences into smaller, focused statements.

Answer: We appreciate the reviewer’s feedback and have revised the abstract to improve clarity, readability, and conciseness. We have removed redundant commas, clarified statements such as “improving the security and resilience of healthcare data”, and rephrased metric descriptions for better flow. Additionally, we have broken down long sentences into smaller, more focused statements to enhance comprehension, particularly for readers unfamiliar with the topic.

Introduction

The introduction is clear w.r.t objectives. It clearly outlines the challenges of existing models and contribution list is also clear. However, grammar mistakes are found at Line 47: "Quality of security and privacy healthcare data concerns" should be rephrased to "Concerns over the quality of security and privacy in healthcare data." Line 55: "A service-reliant authentication approach has been shown to enhance" should specify how it is relevant to IoMT healthcare. Repetition of sentences contextually in line 46, 48, 53, 75). Some references are mentioned without integrating them into the narrative (e.g., Line 55). Contribution 4 (Lines 91–92) overlaps conceptually with Contribution 1 (Lines 85–86). You have explored existing literature to find this problem, which is a contribution. This can be mentioned in contribution list. The term "Machine Learning" (ML) has not been defined in the text before the abbreviation "ML" is used. Rewrite the novelty of your research solution by adding impact and clarity.

Answer: We appreciate the reviewer’s suggestions and have incorporated the necessary revisions to enhance clarity and impact. The phrase "Quality of security and privacy healthcare data concerns" has been rephrased as "Concerns over the quality of security and privacy in healthcare data," ensuring better readability. We have specified the relevance of the service-reliant authentication approach to IoMT healthcare environments, demonstrating its effectiveness in enhan

---

## [Decision Letter · Decision Letter 1]

14 Mar 2025

HCAP: Hybrid Cyber Attack Prediction Model for Securing Healthcare Applications

PONE-D-24-51339R1

Dear Dr. Jamil Abedalrahim Jamil Alsayaydeh,

We’re pleased to inform you that your manuscript has been judged scientifically suitable for publication and will be formally accepted for publication once it meets all outstanding technical requirements.

Kind regards,

mamoona humayun

Academic Editor

PLOS ONE

Additional Editor Comments (optional):

Reviewers' comments:

Reviewer's Responses to Questions

**Comments to the Author**

If the authors have adequately addressed your comments raised in a previous round of review and you feel that this manuscript is now acceptable for publication, you may indicate that here to bypass the “Comments to the Author” section, enter your conflict of interest statement in the “Confidential to Editor” section, and submit your "Accept" recommendation.

Reviewer #1: All comments have been addressed

Reviewer #2: All comments have been addressed

---

## [Editor Report · Acceptance letter]

PONE-D-24-51339R1

PLOS ONE

Dear Dr. Alsayaydeh,

I'm pleased to inform you that your manuscript has been deemed suitable for publication in PLOS ONE. Congratulations! Your manuscript is now being handed over to our production team.

Kind regards,

on behalf of

Dr. mamoona humayun

Academic Editor

PLOS ONE